# Olive Leaf Extract (OLE) as a Novel Antioxidant That Ameliorates the Inflammatory Response in Cystic Fibrosis

**DOI:** 10.3390/cells12131764

**Published:** 2023-07-01

**Authors:** Caterina Allegretta, Graziana Difonzo, Francesco Caponio, Grazia Tamma, Onofrio Laselva

**Affiliations:** 1Department of Clinical and Experimental Medicine, University of Foggia, 71122 Foggia, Italy; caterina.allegretta@unifg.it; 2Department of Soil, Plant and Food Sciences, University of Bari Aldo Moro, 70125 Bari, Italy; graziana.difonzo@uniba.it (G.D.); francesco.caponio@uniba.it (F.C.); 3Department of Bioscience, Biotechnology and Biopharmaceutics, University of Bari Aldo Moro, 70125 Bari, Italy; grazia.tamma@uniba.it

**Keywords:** olive leaf extract, CFTR, Cystic Fibrosis, antioxidant, anti-inflammatory, CFTR modulators

## Abstract

The deletion of phenylalanine at position 508 (F508del) produces a misfolded CFTR protein that is retained in the ER and degraded. The lack of normal CFTR channel activity is associated with chronic infection and inflammation which are the primary causes of declining lung function in Cystic Fibrosis (CF) patients. Moreover, LPS-dependent oxidative stress downregulates CFTR function in airway epithelial cells. Olive leaf extract (OLE) is used in traditional medicine for its effects, including anti-oxidant and anti-inflammatory ones. We found that OLE decreased the intracellular ROS levels in a dose–response manner in CFBE cells. Moreover, OLE attenuates the inflammatory response to LPS or IL-1β/TNFα stimulation, mimicking the infection and inflammatory status of CF patients, in CFBE and primary nasal epithelial (HNE) cells. Furthermore, we demonstrated that OLE restored the LPS-mediated decrease of Trikfafta^TM^-dependent F508del-CFTR function in CFBE and HNE cultures. These findings provide strong evidence of OLE to prevent redox imbalance and inflammation that can cause chronic lung damage by enhancing the antioxidant activity and attenuating inflammation in CF airway epithelial cells. Additionally, OLE might be used in combination with CFTR modulators therapy to improve their efficacy in CF patients.

## 1. Introduction

Cystic fibrosis (CF) is a multi-system, lethal, autosomal recessive disorder caused by defects in the gene encoding for the CF transmembrane conductance regulator (CFTR), an ATP-gated channel for the efflux of chloride and bicarbonate anions mainly expressed in the apical membrane of epithelial cells [1]. The absence or dysfunction of the CFTR protein results in the failure of chloride secretion and sodium hyperabsorption at the apical airway surface, leading to the dehydration of the airway surface, impaired mucociliary clearance, and the accumulation of viscous mucus at the epithelial surface [2]. As a result, CF patients are prone to contracting bacterial lung infections with opportunistic pathogens, especially *Pseudomonas aeruginosa* [3]. Chronic pulmonary infection leads to an excessive immune response, mainly mediated by activated leukocytes, which produce large amounts of reactive oxygen species (ROS) [4,5].

More than 2000 mutations have been identified in CFTR, though not all have been linked with CFTR dysfunction and CF (www.genet.sickkids.on.ca, accessed on 3 May 2023; www.CFTR2.org, accessed on 3 May 2023). The most common mutation, the deletion of phenylalanine at position 508, induces misfolding of the protein that is retained in the ER and degraded by proteasomal pathways. CFTR correctors are pharmacological compounds that rescue the CFTR to the cell surface [6]. Lumacaftor (or VX-809) has been shown to rescue F508del-CFTR function to approximately 15% of normal channel activity in human bronchial epithelial cells treated in combination with the potentiator Ivacaftor (VX-770) [7]. The combination Lumacaftor-Ivacaftor has been approved by the FDA as Orkambi^®^ for patients bearing F508del mutation, and then the related Tezacaftor (VX-661)-Ivacaftor combination has been approved as Symdeko^®^. Recently, the FDA approved the next-generation CFTR modulator (Trikafta^®^), which consists of the potentiator ivacaftor and two correctors molecules, elexacaftor (VX-445) and tezacaftor, for patients carrying at least one F508del allele [8].

It has been well established that dysfunction of the CFTR protein determines a redox imbalance in epithelial cells with the consequent abnormal generation of reactive oxygen species (ROS) [9,10] and a dysregulated inflammatory response, before any infections, by providing a favorable environment for tissue damage and chronic infection [11]. After bacterial colonization, there is an exaggerated activation of the host immune system with the release of proteases, ROS and reactive nitrogen species (RNS), and proinflammatory chemokines, mainly interleukin (IL)-8, by epithelial and inflammatory cells causing tissue damage [12] and lung-function decline in patients affected by CF [13]. Despite recognition of the impact of airway inflammation in the progression of lung disease, few therapies with anti-inflammatory effects have been found beneficial to CF patients (i.e., Ibuprofen). However, despite this, it was found to slow disease progression; the adoption of Ibuprofen has been limited due to dosing issues and serious side effects associated with long-term administration [14,15]. However, it has not yet elucidated the potential anti-inflammatory and anti-oxidative activity of CFTR modulators in CF.

Olive leaves are widely used in natural medicine with beneficial effects on health care due to their bioactive properties [16]. Indeed, Olive leaf extract (OLE) contains large amounts of polyphenolic compounds that include the secoiridoids (namely oleuropein) and flavonoids (~2% of olive leaf polyphenols) with antioxidant, anti-inflammatory and antimicrobial properties [17]. It has been demonstrated that olive left biophenols are able to scavenge ROS and reactive nitrogen species (RNS) [18,19] by activating transcription factors such as the nuclear factor erythroid 2–related factor 2 (Nrf2). Upon oxidative stress, the transcription factor Nrf2 translocates to the nucleus and binds to the antioxidant responsive element (ARE) in the promoter region of genes, encoding enzymes involved in the antioxidant response, i.e., NAD(P)H quinone-oxidoreductase 1 (NQO1), heme-oxyganese-1(HO-1), superoxide dismutase (SOD), glutamate-cysteine ligase (GCL). Moreover, Nrf2 also exhibits anti-inflammatory activity by inhibiting NF-kB signaling, thus inducing the downregulation of pro-inflammatory cytokines (i.e., interleukin (IL)-1β, tumor necrosis factor (TNF-α) gene expression) [20].

In the present study, we demonstrated for the first time that OLE exhibits both antioxidant and anti-inflammatory properties in human bronchial epithelial cells affected by CF (CFBE) at the basal level and stimulated by infection and inflammation stimuli. Moreover, we further demonstrated the anti-inflammatory activity of OLE in primary nasal epithelial cells from CF patients bearing the F508del mutation. Lastly, we demonstrated that OLE restored Trikafta-mediated F508del-CFTR function in CFBE cells and primary nasal epithelial cells stimulated by LPS from *P. aeruginosa*.

## 2. Materials and Methods

### 2.1. Cell Culture

CFBE41o- cells stably expressing F508del-CFTR (CFBE) were obtained from Dr. Gruenert (University of California, San Francisco, CA, USA) and maintained in MEM (Euroclone, Milan, Italy), supplemented with 10% FBS (Corning, New York, NY, USA), L-glutamine (Euroclone), Penicillin/streptomycin (Euroclone) and 2 μg/mL puromycin (Sigma-Aldrich, Milan, Italy) at 37 °C with 5% CO_2_, as previously described [21].

Primary nasal epithelial cells from CF patients homozygous for F508del mutation were obtained from the Canadian Program for Individualized CF Therapy (CFIT) (https://lab.research.sickkids.ca/cfit, accessed on 3 May 2023). Nasal epithelial cells were cultured as previously described [22].

### 2.2. Olive Leaf Extract (OLE) Production

The olive leaf extract was produced according to Difonzo et al. [23]. The olive leaves were dried at 120 °C for 8 min in a ventilated oven (Argolab, Carpi, Italy) and grounded with a blender (Waring-Commercial, Torrington, CT, USA). Milli-Q water was used as extraction solvent in a ratio 1/20 (*w*/*v*). The extraction process was ultrasound-assisted (CEIA, Viciomaggio, Italy) and the extract was filtered through Whatman filter paper (GE Healthcare, Milan, Italy), freeze-dried (BUCHI, Flawil, Switzerland, LyovaporTM L-200), and stored at −20 °C.

The OLE was characterized for total phenol content and antioxidant activity by Folin–Ciocalteu and ABTS assays (119 mg GAE/g and 687 µmol TE/g). The phenolic profile was determined according to Difonzo et al. [24] and the oleuropein content (85 mg/g) was determined by HPLC-DAD and external calibration curve with the relative standard.

### 2.3. Cytotoxicity MTT Assay

CFBE cells (1 × 10^4^ cells/well) were seeded in a 96-well plate (Corning, New York, NY, USA) and cultured in the completed medium for 2 days. Furthermore, 24 h prior to the experiment, the cells were treated with Olive Leaf Extract (OLE) at 0.03, 0.06, 0.12 and 0.24 mg/mL concentrations. Cell viability was then evaluated with the MTT (3-(4,5-dimethylthiazol-2-yl)-2,5 diphenyl tetrazolium bromide) assay by measurement of the optical density at 595 nm, as previously described [25]. Untreated cells are considered as 100% and cells treated with 1% of Triton-X100 were used as a positive control.

### 2.4. ROS Measurment

CFBE cells (1 × 10^4^ cells/well) were seeded in a 96-well plate and cultured for 48 h. Then, cells were treated with 1 μg/mL LPS derived from *P. aeruginosa* (Sigma-Aldrich) or 30 ng/mL IL-1β + 30 ng/mL TNF-α +/− OLE at 0.03, 0.06 and 0.12 mg/mL concentrations for 24 h. ROS were detected with 10 µM o5-(and 6)-chloromethyl 20,70-dichlorodihydrofluorescein diacetate acetyl ester (H2DCFDA, Invitrogen, Waltham, MA, USA) by measuring the fluorescence at 37 °C (excitation: 485 nm; emission: 535 nm), as previously reported [25].

### 2.5. RNA Extraction and Quantification (qRT-PCR)

Cells were seeded at the density of 8 × 10^4^ cells/cm^2^ for 24 h in a six-well plate. After 24 h, cells were then co-treated for 4 h at 37 °C with PBS, 1 μg/mL LPS derived from *P. aeruginosa* (Sigma-Aldrich) or 30 ng/mL IL-1β + 30 ng/mL TNF-α +/− OLE at 0.03, 0.06 and 0.12 mg/mL concentrations. Then, CFBE and primary nasal epithelial cells were lysed, and RNA was extracted according to the manufacturer’s protocol (Direct-zol RNA Kits, Zymo Research, Tustin, CA, USA). The quality of total RNA was assessed by the ratios OD260/OD280 and OD260/OD230 using Nano-drop 2000. Total RNA was converted to cDNA with an iSCRIPT cDNA synthesis kit (Biorad, Hercules, CA, USA) and quantitative real-time PCR was performed using Ssfast EvaGreen (Biorad, Hercules, CA, USA) and normalized to GADPH. The primer sets used for amplification are provided in Table 1 [25].

### 2.6. CFTR Channel Function

CFTR-mediated membrane depolarization was measured as previously described [26,27]. Briefly, CFBE and primary nasal epithelial cells (3 × 10^4^ cells/well) were seeded on 96-well transwells. Cells were treated with 0.1% DMSO, 3 μM VX-661 + 3 μM VX-445 +/− 10 μg/mL LPS +/− 0.12 mg/mL OLE for 24 h at 37 °C. Cells were then loaded with blue membrane potential dye and dissolved in a chloride-free buffer for 30 min at 37 °C [28]. The plate was read in a fluorescence plate reader (FilterMax F5, Molecular Devices) at 37 °C and after reading the baseline fluorescence, CFTR was stimulated by 10 μM forskolin (FSK) + 1 μM VX-770 (Selleck Chemicals, Houston, TX, USA). CFTR-mediated depolarization of the membrane was detected as an increase in fluorescence and repolarization or hyperpolarization as a decrease [29]. Lastly, the CFTR inhibitor (CFTRInh-172, 10 μM) was added to deactivate CFTR. The peak changes in fluorescence to CFTR agonists were normalized relative to fluorescence immediately before agonist (FSK + VX-770) addition. The max activation expressed as a % was calculated by the difference between the peak of maximal CFTR activation and the last point of baseline [30].

### 2.7. Statistical Analysis

All the data are represented as mean ± SEM of at least three independent replicates. GraphPad 8.0 software was used for all statistical analysis. The paired two-tailed *t*-test or one-way ANOVA were conducted as appropriate with a significant level *p* < 0.05. Data with multiple comparison were assessed using Turkey’s multiple comparison test with α = 0.05.

## 3. Results

### 3.1. Effect of Olive Leaf Extract on the Cell Viability of CFBE Cells

We first employed the MTT assay to investigate the potential cytotoxicity of OLE on CFBE cells. Cells were treated with increasing concentrations (0.03; 0.06; 0.12 and 0.24 mg/mL) of OLE for 24 h (Figure 1). While concentrations from 0.03 to 0.12 mg/mL do not affect cell viability compared to untreated control, OLE at 0.24 mg/mL significantly reduced the viability of CFBE; therefore, this concentration (0.24 mg/mL) was excluded for further studies.

### 3.2. OLE Reduces Oxidative Stress (ROS) in CFBE Cells

To evaluate the antioxidant activity of OLE in CFBE cells, the ROS content was evaluated by a ROS-sensitive fluorescent probe (H2DCFDA). A significant reduction of intracellular levels of ROS was observed in untreated CFBE cells only at 0.12 mg/mL concentration (Figure 2). As expected, the intracellular levels of ROS were increased after infection (LPS-dependent) or inflammation (IL-1β/TNFα) stimuli (Figure 2). Interestingly, OLE at 0.12 mg/mL significantly reduced the ROS levels with all stimuli, similarly to the level measured under basal condition (control cells). Therefore, these studies suggested that OLE has antioxidant activity in CF bronchial cells.

### 3.3. Anti-Inflammatory Activity of OLE

To further investigate the anti-inflammatory activity of OLE in CF, we measured the mRNA expression levels of pro-inflammatory cytokines (IL-1β, IL-6, IL-8 and TNFα) by RT-PCR. We first tested the effect of OLE on CFBE cells at basal condition. The mRNA expression levels of IL-1β, IL-6, IL-8 and TNFα decreased in a OLE dose-dependent manner (Figure 3A). Then, we tested the effect of OLE in CFBE cells co-treated with LPS from *P. aeruginosa* or IL-1β/TNFα for 4 h to mimic the infection or inflammatory milieu of CF airways. Interestingly, OLE significantly reduced the mRNA levels of pro-inflammatory cytokines under infection/inflammation stimuli (Figure 3B,C). These data strongly suggested that OLE inhibits the expression levels of cytokines associated with CF airways inflammation in a CF cell line.

We recently demonstrated that LPS treatment significantly reduced F508del-CFTR function, rescued by the triple combination VX-445 + VX-661 + VX-770 (Trikafta^®^) [25]. Therefore, we further investigated the effect of OLE on F508del-CFTR function rescued by Trikafta^®^ under LPS treatment. As shown in Figure 4A,B, OLE restored the Trikafta^®^-mediated F508del-CFTR activity in CFBE cells stimulated with LPS to that of non-LPS treated. Moreover, OLE treatment did not exhibit correction activity on F508del-CFTR function. In Figure 4C,D, we demonstrated that LPS treatment significantly reduced the VX-661 + VX-445-mediated F508del-CFTR, which was rescued by OLE treatment.

Together, these findings prompted us to validate the multi-task activity of OLE in a more relevant cell model used as a preclinical tool for personalized medicine in CF. Therefore, we generated primary nasal epithelial cultures from three CF patients homozygous for the F508del mutation. As shown in Figure 5A, LPS treatment increased the mRNA expression of pro-inflammatory cytokines (IL-1β, IL-6, IL-8 and TNFα). However, OLE treatment significantly reduced the pro-inflammatory cytokines mRNA levels to the basal level.

We next studied the effect of OLE on Trikafta^®^ efficacy to rescue F508del-CFTR under LPS treatment. In Figure 5B,C, we showed that OLE restored Trikafta^®^-mediated rescue of the F508del-CFTR function treated with LPS in primary nasal epithelial cells from three CF patients.

To investigate the mechanism of action of OLE in airway epithelial cells, we studied the expression by RT-PCR of Nrf2, HO-1, NQO1 and NF-kB in both CFBE and primary nasal epithelial cells after treatment with LPS in the presence or absence of OLE. Interestingly, OLE significantly increased Nrf2, HO-1 and NQO1 and reduced the nuclear NF-kB dimers (p50/p65) mRNA expression levels (Figure 6).

## 4. Discussion

Our study provides novel data on the positive effect of olive leaf extract on the human airway epithelial cells from Cystic Fibrosis patients. We found that OLE reduced ROS production and suppressed the induction of pro-inflammatory cytokines by regulating Nrf2 and NF-kB pathways.

OLE provides multiple beneficial effects for its multi-target activities, such as: antioxidation, anti-inflammation, anti-hypertensive, hypoglycemic, cardioprotection, autophagy inducing, anti-cancerous and anti-amyloid aggregation [31,32,33,34,35,36,37,38,39]. Therefore, olive leaf consumption has been associated with various health benefits, including protection against neurodegenerative diseases (i.e., Multiple Sclerosis, Alzheimer and Parkinson), metabolic syndromes (i.e., Obesity, hyperglycemia, diabetes mellitus), cardiovascular diseases (Arterial hypertension and atherosclerosis) and inflammatory bowel diseases (i.e., Ulcerative colitis) in preclinical models and in humans [40,41,42,43,44,45].

OLE has been marketed as a natural supplement for multiple health benefits, due to the different compounds that bring about its anti-oxidant and anti-inflammatory activity (i.e., Oleuropein) [46]. It has been demonstrated that Oleuropein reduced the pro-inflammatory cytokines (IL-1β, IL-6 and IL-8) under LPS-stimulated conditions by downregulating the arachidonic acid and NF-kB pathways [47,48,49,50]. Moreover, it has been demonstrated that OLE exhibits its antioxidant activity by decreasing the ROS content of about 40% in human bronchial epithelial NCl-H292 cells [51]. Our aim was to determine the anti-inflammatory effects of OLE in CF airways cells. We found a significant inhibition of pro-inflammatory cytokine gene expression in CFBE cells at a basal condition (Figure 3A). We then tested the anti-inflammatory activity of OLE under infection or inflammatory condition to mimic in vivo conditions. Therefore, we used two different stimuli such as LPS from *P. aeruginosa* to mimic the CF lung infection [52] and IL-1β/TNFα to mimic the inflammatory milieu [53]. We found that OLE reduced the pro-inflammatory cytokines gene expression even under infection/inflammation stimuli conditions. Interestingly, the anti-inflammatory properties of OLE was confirmed in primary CF nasal epithelial cultures under LPS treatment (Figure 5A). Moreover, we found that OLE reduced the NF-kB mRNA levels of both nuclear dimers (p50 and p60) in both CFBE and primary nasal epithelial cells under LPS stimulation. Since previous in vitro and in vivo studies demonstrated that the anti-inflammatory activity of OLE is via the regulation of NF-kB, due to the presence of phenolic compounds in the extract [54,55], OLE may reduce the pro-inflammatory cytokines in CFBE and primary nasal epithelial cells due the presence of flavonoids and phenolic secoiridoids identified in the OLE, which is well known their anti-inflammatory activities [56,57,58].

It has been demonstrated that OLE acts as an Nrf2 inducer, which correlates with the increased expression of HMOX-1 and SOD, and decreased iNOS expression and subsequent NO production [59]. Nrf2 is a regulator of cellular resistance to oxidant stress by increasing the expression of antioxidant response element-dependent genes encoding enzymes involving in the antioxidant response (i.e., Glutathione-S-transferasi (GST), Superoxide Dismutase (SOD), NAD(P)H quinone-oxidoreductase 1 (NQO1) and heme-oxygenase-1 (HMOX-1)) [60]. Interestingly, it is well demonstrated that activation of Nrf2, in addition to attenuate oxidative stress, suppress NF-kB, iNOS (produce reactive NO species with pro-inflammatory effects) and pro-inflammatory cytokines [61,62]. We demonstrated that OLE by up-regulating the Nrf2/HO-1/NQO1 signaling (Figure 6), may boost the antioxidant defenses and protect against oxidative stress in both CFBE and primary nasal epithelial cells from CF patients.

All together, we assumed that the observed antioxidant and anti-inflammatory effect of OLE might be mediated via its ability to up-regulate Nrf2 pathway. Indeed, OLE induced the up-regulation of Nrf2 expression with the consequent induction of HO-1 and NQO1 and attenuate the oxidative stress. Moreover, OLE reduced the expression of NF-kB and pro-inflammatory cytokines.

It has previously been demonstrated that infection with *P. aeruginosa* bacteria reduce the functional expression of WT-CFTR or Orkambi^®^-rescued F508del-CFTR in CFBE and primary epithelial cells [63,64,65]. This observation may explain why Orkambi^®^ has a modest effect in clinical trials [66]. Thus, it is likely that the reduction we observed in F508del-CFTR function expression rescued by Trikafta^®^ in CFBE and primary nasal epithelial cells treated with LPS were due to the *P. aeruginosa* infection (Figure 4 and Figure 5). These data indicate that *P. aeruginosa* exoproducts may affect CFTR synthesis, degradation and trafficking to the cell membrane [64]. However, the mechanism whereby F508del-CFTR rescue is affected by *P. aeruginosa* is still unknown. Interestingly, we also found that OLE restores Trikafta^®^ rescue of F508del-CFTR function in the CFBE cell line and primary nasal cells treated with LPS (Figure 4 and Figure 5). Since it has been demonstrated that oxidative stress in CF airways has a direct deleterious effect on CFTR function and amplification of lung inflammation [67,68], we speculate that OLE restored Trikafta^®^-mediated F508del-CFTR function under LPS treatment through its antioxidant activity. On the other hand, the combination of OLE and CFTR modulators did not have a synergistic effect (data not shown).

The present study is the first to show that OLE exhibits anti-oxidative and anti-inflammatory properties in the CF bronchial cell line and CF primary nasal epithelial cells. Moreover, OLE is an effective and economic natural herb that seems to improve the efficacy of CFTR modulators therapies under infection stimuli, which could provide a clear benefit to CF patients with *P. aeruginosa* colonization. Together, these data suggest OLE as a potential therapeutic approach for treating a harshly inflamed CF lung. However, further investigations are warranted to explore the clinical application of OLE.

## Figures and Tables

**Figure 1 cells-12-01764-f001:**
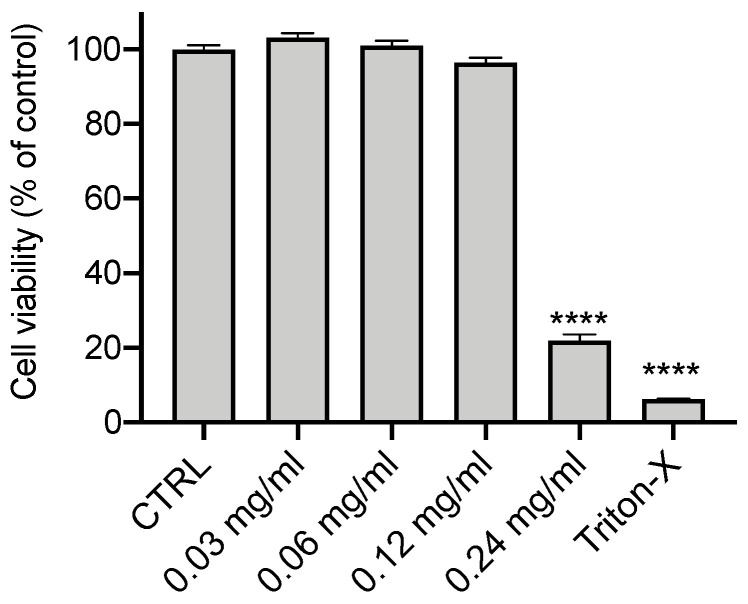
Cytotoxicity of OLE. CFBE cells were treated with OLE for 24 h at the indicated concentrations of 0.03, 0.06, 0.12 and 0.24 mg/mL. Control (CTRL) cells were untreated (100% of vitality) and 1% Triton X-100 was used as a positive control. Cells were then assayed for vitality by the MTT assay. Data represent the mean ± SEM (*n* = 12). **** *p* < 0.0001.

**Figure 2 cells-12-01764-f002:**
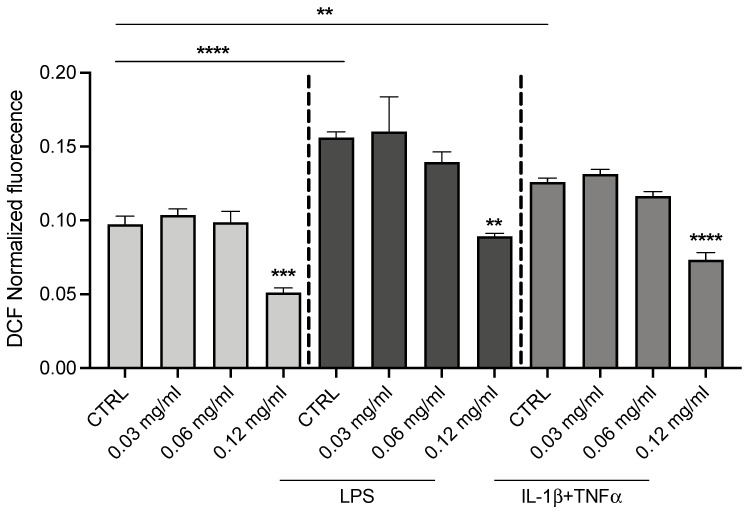
Treatment with OLE reduces ROS production in CFBE cells. ROS content was measured using DCF fluorescence in CFBE cells stimulated by 1 µg/mL of LPS or 30 ng/mL IL-1b + 30 ng/mL TNFα in the presence or absence of the OLE (0.03, 0.06 and 0.12 mg/mL). Data represent the mean ± SEM (*n* = 4). ** *p* < 0.01; *** *p* < 0.001; **** *p* < 0.0001.

**Figure 3 cells-12-01764-f003:**
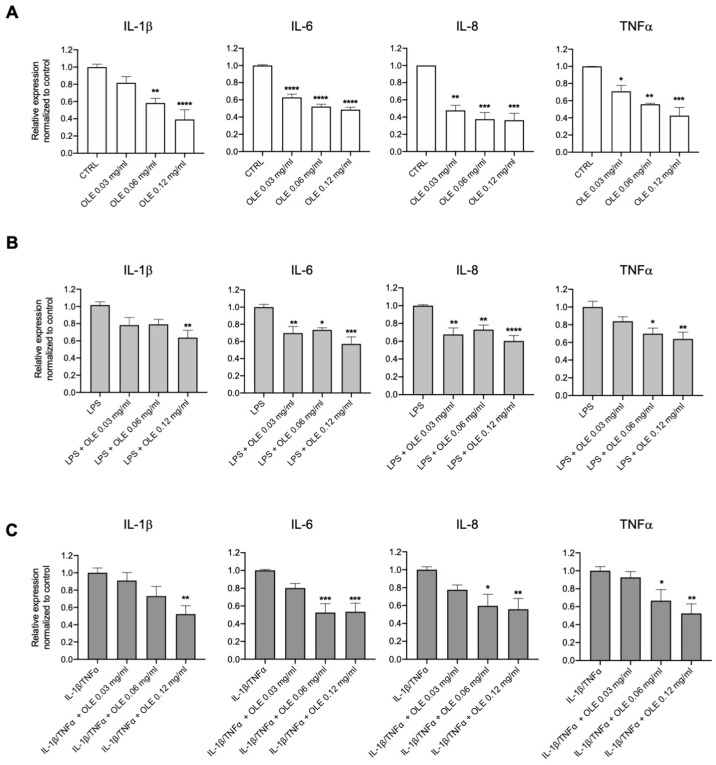
OLE treatment reduces the expression of cytokines stimulated by LPS and IL-1β and TNFα in CFBE cells. CFBE cells were treated with (**A**) Medium, (**B**) 1 µg/mL of LPS or (**C**) 30 ng/mL IL-1β + 30 ng/mL TNFα +/− OLE at concentrations of 0.03, 0.06 and 0.12 mg/mL for 4 h. IL-1β, IL-6, IL-8 and TNFα were quantified by qRT-PCR and normalized to GADPH as housekeeping gene and then to control untreated cells. Data represent the mean ± SEM (*n* = 3). * *p* < 0.05; ** *p* < 0.01; *** *p* < 0.001; **** *p* < 0.0001.

**Figure 4 cells-12-01764-f004:**
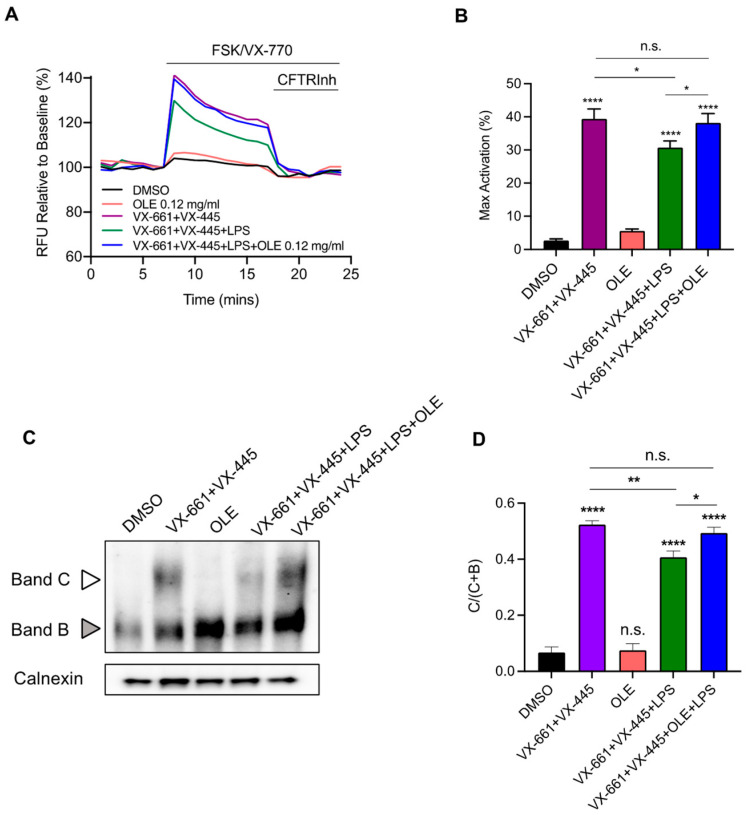
OLE treatment restored Trikafta-mediated F508del-CFTR function in CFBE cells stimulated by LPS. (**A**) Representative traces of F508del-CFTR function using membrane depolarized assay. F508del-CFTR CFBE cells were treated with 0.1% DMSO, 0.12 mg/mL OLE, 3 μM VX-661 + 3 μM VX-445, 3 μM VX-661 + 3 μM VX-445 + 0.12 mg/mL OLE +/− 1 μg/mL of LPS for 24 h. (**B**) Bar graphs show the mean (±SEM) of maximal activation of CFTR after stimulation by 10 μM forskolin (FSK) + 1 μM VX-770 (*n* = 6). (**C**) Immunoblots of steady-state expression F508del-CFTR in CFBE cells following the indicated treatments. (**D**) Bars represent the mean (±SEM) of the ratio C/C + B (*n* = 3). * *p* < 0.05; ** *p* < 0.01; **** *p* < 0.0001.

**Figure 5 cells-12-01764-f005:**
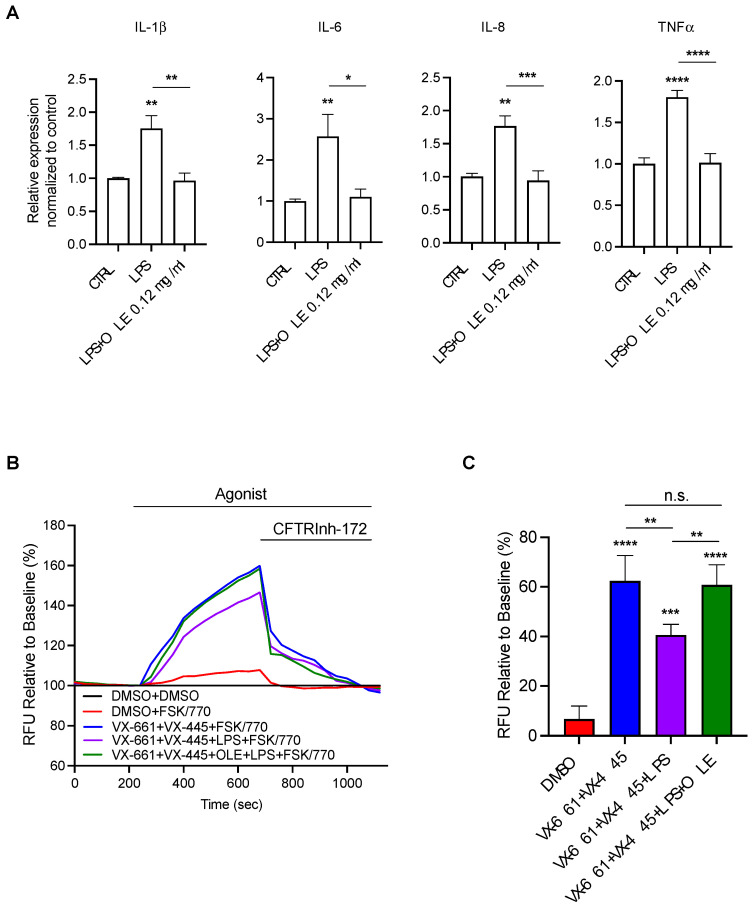
OLE treatment reduces the pro-inflammatory cytokines and restored Trikafta-mediated F508del-CFTR function in CF primary nasal epithelial cells stimulated by LPS. (**A**) Primary nasal epithelial cells from 4 CF patients (F508del/F508del) were treated with 1 µg/mL of LPS +/− OLE at 0.12 mg/mL for 4 h. IL-1β, IL-6, IL-8 and TNFα were quantified by qRT-PCR and normalized to GADPH as housekeeping gene and then to control untreated cells. Data represent the mean ± SEM (*n* = 8). * *p* < 0.05; ** *p* < 0.01; *** *p* < 0.001; **** *p* < 0.0001 (**B**) Representative traces of F508del-CFTR function in primary nasal epithelial cells from 3 CF patients homozygous for F508del mutation, using membrane depolarized assay. Cells were treated with 0.1% DMSO, 3 μM VX-661 + 3 μM VX-445, 3 μM VX-661 + 3 μM VX-445 + 1 μg/mL of LPS +/− 0.12 mg/mL OLE for 24 h. (**C**) Bar graphs show the mean (±SEM) of maximal activation of CFTR after stimulation by 10 μM forskolin (FSK) + 1 μM VX-770 (*n* = 6). ** *p* < 0.01; *** *p* < 0.001; **** *p* < 0.0001.

**Figure 6 cells-12-01764-f006:**
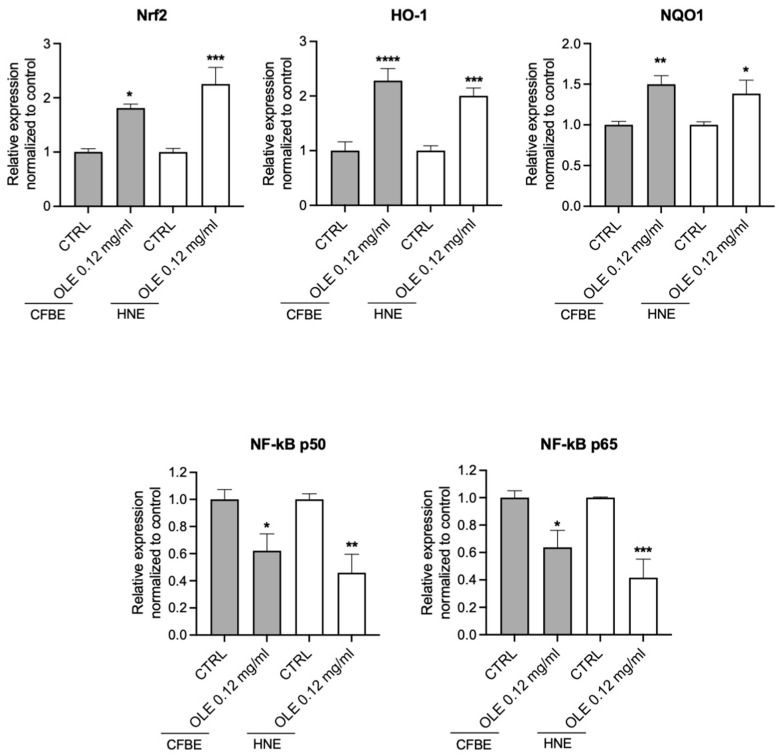
OLE treatment upregulates Nrf2/HO-1/NQO1 and down-regulates NF-kB signaling in CFBE and primary nasal epithelial cells stimulated by LPS. F508del-CFTR CFBE and primary nasal epithelial cells from 3 CF patients homozygous for F508del mutation were treated with 1 µg/mL of LPS +/− OLE at 0.12 mg/mL for 4 h. Nrf2, HO-1, NQO1, NF-kB p50 and NF-kB p65 were quantified by qRT-PCR and normalized to GADPH as housekeeping gene and then to control untreated cells. Data represent the mean ± SEM (*n* = 4 for CFBE and *n* = 6 for primary nasal epithelial cells). * *p* < 0.05; ** *p* < 0.01; *** *p* < 0.001; **** *p* < 0.0001.

**Table 1 cells-12-01764-t001:** Primer pairs used in real-time PCR analysis.

Gene	Forward Primer (5′ → 3′)	Reverse Primer (5′ → 3′)
IL-1β	TTACAGTGGCAATGAGGATGAC	TGTAGTGGTGGTCGGAGATTC
IL-6	CGGTACATCCTCGACGGC	CTTGTTACATGTCTCCTTTCTCAGG
IL-8	GACCACACTGCGCCAACA	GCTCTCTTCCATCAGAAAGTTACATAATTT
TNFα	GGACCTCTCTCTAATCAGCCCTC	TCGAGAAGATGATCTGACTGCC
Nrf2	CACATCCAGTCAGAAACCAGTGG	GGAATGTCTGCGCCAAAAGCT
NF-kB p50	CTGGTGATCGTGGAACAGCC	CAGAGCCTGCTGTCTTGTCC
NF-kB p65	ATGCGCTTCCGCTACAAGTG	ACAATGGCCACTTGTCGGTG
HO-1	ATGGAGCGTCCGCAACCCGACAG	TCACATGGCATAAAGCCCTACAG
NQO1	GAAGAGCACTGATCGTACTGGC	GGGTCCTTCAGTTTACCTGTGAT
GAPDH	CAAGAGCACAAGAGGAAGAGAG	CTACATGGCAACTGTGAGGAG

## Data Availability

The data that support the findings of this study are available from the corresponding author upon reasonable request.

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
