# Peer review of "Olive Leaf Extract (OLE) as a Novel Antioxidant That Ameliorates the Inflammatory Response in Cystic Fibrosis"

_cells, 2023, doi:10.3390/cells12131764_

Round 1
Reviewer 1 Report
The authors provide strong evidence of Olive Leaf Extract (OLE) prevents redox imbalance and inflammation that cause chronic lung damage by enhancing the antioxidant activity and attenuating inflammation in CF airway epithelial cells. They also show that OLE may restore Trikafta® rescue of F508del-CFTR function in CFBE cell line and primary nasal cells treated
with LPS. The latter result although it suggests that OLE might be used in combination with CFTR modulators therapy to improve their efficacy in CF patients, the authors claim, but not show, that the combination of OLE and CFTR modulators did not have a synergistic effect.
This is an interesting paper where a natural produced substance may help to treat certain cases of CF. Further analysis of the potential cooperative effect of CTFR modulators with OLE will be of interest.
Reviewer 2 Report
The authors showed interesting results, including that OLE inhibited ROS production in CF airway epithelial cells, inhibited inflammation, and restored F508del-CFTR correction by VX-661 + VX-445 that was reduced by LPS. However, it contains several deficiencies that must meet the journal's quality standards.
In Figure 2, LPS-induced ROS was significantly reduced only when CFBE cells were treated with 0.12 mg/ml OLE, whereas in Figure 3, lower concentrations of 0.03 or 0.06 mg/ml OLE significantly reduced cytokine expression. A clear explanation is needed as to why cytokine expression is significantly reduced even at low concentrations of OLE.
In Figure 4, to clearly see the effect of OLE on the LPS-induced reduction of F508del-CFTR correction by VX-661 + VX-445 in CF airway epithelial cells, it is necessary to demonstrate changes in protein expression levels through immunoblot analysis.
In Figure 6, changes in p50 and p65 mRNA expression in relation to NF-kB activity are difficult to interpret, so immunoblot analysis for p-p65 is required to clearly confirm the effect of OLE on NK-kB activity.
There are many typing errors in images and text that need to be fixed.
Round 2
Reviewer 2 Report
I am satisfied with the author’s responses to my previous comments.